# STORY2SCREEN: MULTIMODAL STORY CUSTOMIZATION FOR LONG CONSISTENT VISUAL SEQUENCES

## ABSTRACT

Multimodal Story Customization aims to generate coherent story flows while conditioning on both textual descriptions and reference identity images. While recent progress in story generation has shown promising results, most existing approaches rely on text-only inputs, with a few works incorporating character identity cues (e.g., facial ID) but lacking broader multimodal conditioning. This limited reliance makes it difficult to jointly preserve consistency of characters, scenes, and textual details across frames, thereby constraining the applicability of these approaches in practical domains such as filmmaking, advertising, and storytelling. In this work, we introduce Story2Screen, a multimodal framework that integrates free-form description with character and background references to enable coherent and customizable story generation. To enhance cinematic diversity, we introduce shot-type control via parameter-efficient prompt tuning on movie data, enabling the model to generate sequences that more faithfully reflect real-world cinematic grammar. To comprehensively evaluate our framework, we establish two new benchmarks, MSB and $M^2SB$, which assess multimodal story customization from the perspectives of character/scene consistency, text–visual alignment, and shot-type control. Extensive experiments demonstrate Story2Screen achieves improved consistency and cinematic diversity compared to existing methods.

## 1 INTRODUCTION

Recent advancements in text-to-video (T2V) (Kondratyuk et al., 2023; Liu et al., 2025a) and text-and-image-to-video (TI2V) (Wan et al., 2025) Transformers have shown notable progress in generating compelling clips. However, producing longer videos with consistent characters, coherent scenes, and structured narratives remains a significant challenge. This limitation constrains their applicability in creative domains such as filmmaking and advertising. A central difficulty lies in maintaining long-term consistency, since the computational and memory cost of Transformer self-attention scales quadratically with temporal length. Recent efforts (Chen et al., 2024; Wang et al., 2025a; Dalal et al., 2025) have extended video duration to the minute scale, however, preserving long-term consistency remains an open challenge. Under these limitations, research (Zhao et al., 2025; He et al., 2025) has proposed pipelines to extend the video, where T2I models generate keyframes that are then expanded into video clips via I2V or TI2V models. Concatenating these clips enables longer video generation, making keyframe coherence critical as structural anchors for storytelling. This naturally motivates the task of Multimodal Story Customization (MSC), which extends beyond text-only generation by conditioning on both text prompt and reference images. Unlike producing isolated frames from T2I or TI2I models, MSC aims to generate sequences that preserve character identity, temporal consistency, and maintain narrative coherence.

Existing approaches to story generation (Tewel et al., 2024; Zhou et al., 2024; Wang et al., 2025b) are training-free methods that build on text-to-image diffusion models but cannot utilize reference images for character- or scene-specific customization. Other methods (Liu et al., 2025b; Zhao et al., 2025; Ma et al., 2025) have focused their consistency objectives on foreground consistency (e.g., character faces), while accessories and background continuity are often ignored. To address this, CharaConsist (Wang et al., 2025b) proposes point-tracking attention to improve scene consistency. However, without explicit conditioning on character and scene reference images, story customization and long-term consistency remain difficult. Therefore, we focus on MSC, conditioning on both textual descriptions and visual references to generate coherent story sequences, as shown in Fig. 1.

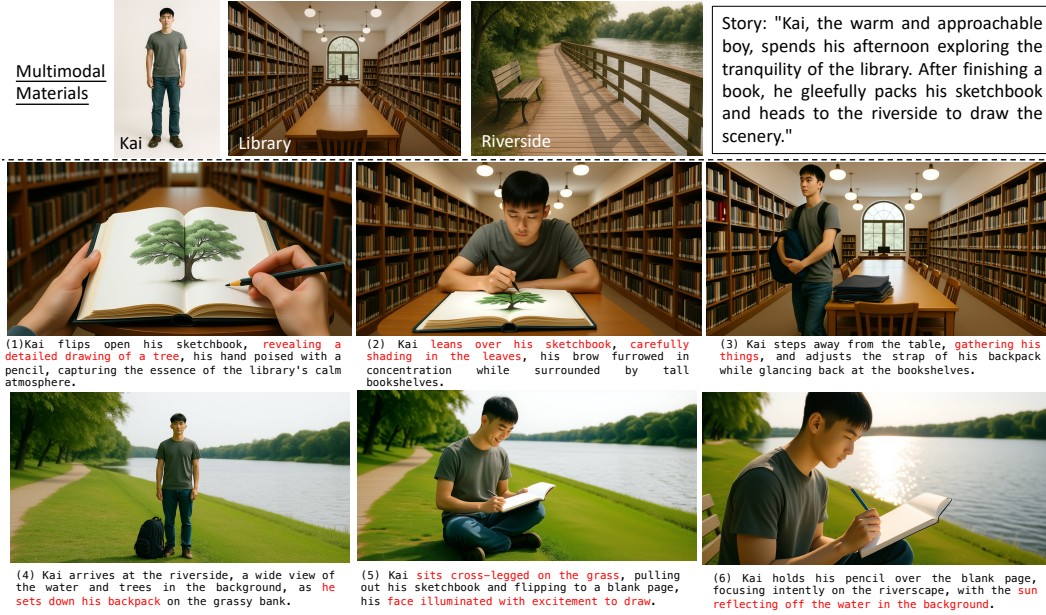

Story: "Kai, the warm and approachable boy, spends his afternoon exploring the tranquility of the library. After finishing a book, he gleefully packs his sketchbook and heads to the riverside to draw the scenery."

(1) Kai flips open his sketchbook, revealing a detailed drawing of a tree, his hand poised with a pencil, capturing the essence of the library's calm atmosphere.

(2) Kai leans over his sketchbook, carefully shading in the leaves, his brow furrowed in concentration while surrounded by tall bookshelves.

(3) Kai steps away from the table, gathering his things, and adjusts the strap of his backpack while glancing back at the bookshelves.

(4) Kai arrives at the riverside, a wide view of the water and trees in the background, as he sets down his backpack on the grassy bank.

(5) Kai sits cross-legged on the grass, pulling out his sketchbook and flipping to a blank page, his face illuminated with excitement to draw.

(6) Kai holds his pencil over the blank page, focusing intently on the riverscape, with the sun reflecting off the water in the background.

Figure 1: Story2Screen can generate (1) consistent keyframes with different shot types based on the multimodal materials. (2) coherent story scenes that reference multiple characters and backgrounds.

Table 1: Features Comparison between ConsistFilmer and existing Story Generation Models

| Methods/Features | Consistency | | Reference Images as ID (Customization) | | | Control | |
|---|---|---|---|---|---|---|---|
| | Character | Scene | Single Character | Multiple Characters | Scene/ Background | Text Prompt | Shot-type |
| IP-Adapter (Arxiv'23) | ✓ | ✗ | ✓ | ✗ | ✗ | ✓ | ✗ |
| StoryGen (CVPR'24) | ✓ | ✗ | ✓ | ✗ | ✗ | ✓ | ✗ |
| ConsiStory (SIGGRAPH'24) | ✓ | ✗ | ✗ | ✗ | ✗ | ✓ | ✗ |
| StoryDiffusion (NeurIPS'24) | ✓ | ✗ | ✗ | ✗ | ✗ | ✓ | ✗ |
| Storynizor (AAAI'25) | ✓ | ✗ | ✓ | ✗ | ✗ | ✓ | ✗ |
| StoryWeaver (AAAI'25) | ✓ | ✗ | ✓ | ✓ | ✗ | ✓ | ✗ |
| DreamStory (TPAMI'25) | ✓ | ✗ | ✓ | ✓ | ✗ | ✓ | ✗ |
| CharaConsist (ICCV'25) | ✓ | ✓ | ✗ | ✗ | ✗ | ✓ | ✗ |
| ConsistFilmer (Ours) | ✓ | ✓ | ✓ | ✓ | ✓ | ✓ | ✓ |

We propose Story2Screen, as shown in Fig. 2, a three-stage framework that transforms free-form descriptions into long video sequences. The process begins by generating structured multimodal scripts for each keyframe, including text prompts and reference images (characters and scenes). These materials are then processed by the proposed ConsistFilmer, a multimodal generation model that can receive multiple multimodal scripts to produce consistent keyframes. By incorporating Inner-batch Text Reference (ITR) and Next Keyframe Prediction (NKP), ConsistFilmer ensures that each keyframe evolves smoothly from the previous one while preserving both character identity and scene continuity. Beyond consistency, ConsistFilmer further incorporates shot-type control through prompt tuning with movie data from CMD (Bain et al., 2020). This design enables ConsistFilmer to produce sequences that are not only coherent but cinematic, overcoming the monotony of fixed-view generation in existing methods. Besides, prior works (Zhou et al., 2024; Wang et al., 2025b) on story generation have mainly concentrated on single-character scenarios. In contrast, ConsistFilmer can be extended to multi-subject story customization. We summarize the key differences between ConsistFilmer and existing story generation models in Tab. 1. We further introduce the Multimodal Storyboard Benchmark (MSB) and the Multimodal & Multisubject Storyboard Benchmark (M²SB), two benchmarks designed to evaluate MSC in single- and multi-subject settings. Both benchmarks include diverse identity images, backgrounds, and narrative scripts, providing a unified basis for fair comparison across existing and future methods. Our contributions are summarized as follows:

- We propose **Story2Screen**, a pipeline for long story generation with multimodal customization. To the best of our knowledge, it is the first framework to achieve customization on text prompts, character(s)/scene references, and shot-type, making it flexible for users.

- We introduce **ConsistFilmer**, a unified multimodal generator that employs ITR and NKP to improve consistency, while supporting rich shot-type control.

- We release two new benchmarks, **MSB** and **M²SB**, containing multimodal materials, which provide the standardized evaluation for multimodal story customization.

## 2 RELATED WORK

**Long Video Generation Models.** Long video generation has been a long-standing challenge in the field. Recent works (Chen et al., 2024; Wang et al., 2025a; Dalal et al., 2025) have extended video duration to minutes, surpassing the lengths achievable by models such as Sora (Brooks et al., 2024) and Veo3 (DeepMind, 2025). SEINE (Chen et al., 2024) is designed to reuse the last few frames of a generated video to predict subsequent ones. LingGen (Wang et al., 2025a) enables linear-complexity generation but remains limited to single scenes and slow motion, without capturing complex narratives. TTT-Video (Dalal et al., 2025) produces clips up to one minute long. To further extend video length, MovieDreamer (Zhao et al., 2025) adopts a multi-stage pipeline that first generates keyframes, expands them into short clips via I2V models, and concatenates them into long videos. However, MovieDreamer primarily focuses on facial consistency while neglecting scene and contextual coherence, resulting in inconsistent scenes and reduced narrative expressiveness.

**Unified Multimodal Models.** Recent efforts have explored unified frameworks for multimodal understanding and generation. Seed-X (Ge et al., 2024) introduces multi-granularity modeling to support both image understanding and generation across arbitrary resolutions. Emu3 (Wang et al., 2024) adopts a purely autoregressive paradigm, tokenizing all modalities to unify image and video generation. Janus-Pro (Chen et al., 2025) employs two decoupled image encoders to separate understanding from generation, thereby enhancing semantic comprehension and visual synthesis. Omni-Gen (Xiao et al., 2025) integrates text-to-image, editing, and in-context generation within a rectified-flow framework with decoupled tokenization. While these models demonstrate strong multimodal understanding and image generation, they fall short in maintaining long-term consistency.

**Visual Story Generation.** The goal of story generation is to produce coherent sequences of images that ensure both character and scene consistency while preserving logical continuity throughout the narrative. Prior works have explored this challenge from different perspectives, with early studies focusing mainly on character (foreground) consistency. For instance, ConsiStory (Tewel et al., 2024) is designed to maintain subject identity while aligning with textual descriptions, StoryDiffusion (Zhou et al., 2024) incorporates Consistent Self-Attention to enhance character preservation, and Storynizor (Ma et al., 2025) introduces ID-Injector and ID-Synchronizer modules for identity consistency. Similarly, StoryWeaver (Zhang et al., 2025), DreamStory (He et al., 2025), and 1P1S (Liu et al., 2025b) continue to prioritize character consistency. Although these methods improve character preservation, they remain limited in enforcing scene coherence, often leading to discontinuities. CharaConsist (Wang et al., 2025b) advances this direction by leveraging point-tracking attention and adaptive token merging to align foreground and background, but it still relies solely on text inputs, restricting effective customization. In contrast, our method is the first to condition on a comprehensive set of multimodal materials, including text prompts, foreground subjects, background scenes, and shot-type annotations, providing greater flexibility for user customization, whereas prior approaches can only handle a subset of these conditions.

## 3 METHOD

The goal of Story2Screen is to generate long visual sequences with narrative coherence, character/scene consistency, and cinematic diversity. To achieve this, we design a three-stage framework. First, the Multimodal Generative Model as Director, which structures the narrative into multimodal scripts, which include textual prompts, reference images, and shot-type annotations. Second, ConsistFilmer produces consistent keyframes by integrating Inner-batch Text Reference (ITR), Next Keyframe Prediction (NKP), and shot-type control. Third, Text-and-Image-to-Video (TI2V) Expan-

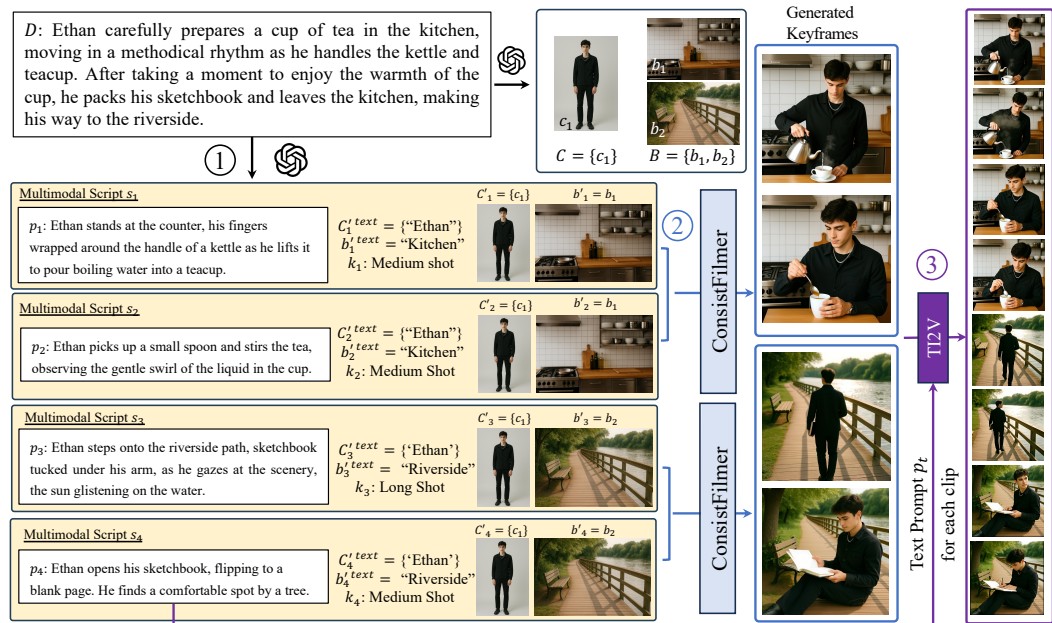

Figure 2: Overview of Story2Screen. (1) Multimodal scripts are generated from text and references; (2) ConsistFilmer produces consistent keyframes; (3) TI2V expands them into long videos.

sion converts script–keyframe pairs from previous stages into short clips, which are then concatenated into long videos. The overview of Story2Screen is illustrated in Fig. 2

## 3.1 MULTIMODAL GENERATIVE MODEL AS DIRECTOR

In this stage, we construct the multimodal materials required for Story2Screen. Given a free-form description $D$, a multimodal generative model (GPT-4o) produces: (1) Text prompts $P = \{p_1, \ldots, p_n\}$, each describing one of $n$ keyframes; and (2) Reference images of story characters $C = \{c_1, \ldots, c_m\}$ and background scenes $B = \{b_1, \ldots, b_o\}$, where $m$ and $o$ denote the numbers of unique characters and backgrounds, respectively. For each keyframe $t$, the script $s_t$ includes a textual prompt $p_t$, reference images $C'_t \subseteq C$ and $b'_t \in B$, and their corresponding textual mentions of character(s) $C'^{\text{text}}_t$ and background $b'^{\text{text}}_t$, and a shot type $k_t \in \mathcal{K}$, where $\mathcal{K}$ denotes the shot-type vocabulary: The multimodal scripts, $s_t = ( p_t, C'_t, C'^{\text{text}}_t, b'_t, b'^{\text{text}}_t, k_t )$ serve as conditions for the next stage, where ConsistFilmer instantiates them into consistent keyframes, enabling characters to move dynamically across scenes, a capability not supported by prior story generation models.

## 3.2 CONSISTFILMER: CONSISTENT KEYFRAME GENERATOR

While multimodal generation models (Wu et al., 2025; Wang et al., 2024) achieve strong results on in-context image generation and editing, they remain limited in producing temporally consistent sequences required for story generation. To overcome this, we propose **ConsistFilmer**, a keyframe generator that enforces both inter-frame alignment with reference images and intra-frame coherence across time (Fig. 3). ConsistFilmer integrates three mechanisms: Inner-batch Text Reference (ITR), Next Keyframe Prediction (NKP), and Shot-type Control.

**Inner-batch Text Reference (ITR).** ITR ensures consistent conditioning within the same scene. Instead of generating each keyframe independently, we jointly encode all textual descriptions associated with a scene. Concretely, for two consecutive scripts $s_{t-1}$ and $s_t$, if $C'^{text}_t = C'^{text}_{t-1}$ and $b'^{text}_t = b'^{text}_{t-1}$, we group them into the same batch and process them with ITR. Formally, given multimodal scripts $S = \{s_1, \ldots, s_n\}$, we concatenate the prompt in each script and obtain hidden states through the autoregressive (AR) model: $H = f_{\text{AR}}([p'_1; \ldots; p'_n])$, where $H = \{h_1, \ldots, h_n\}$

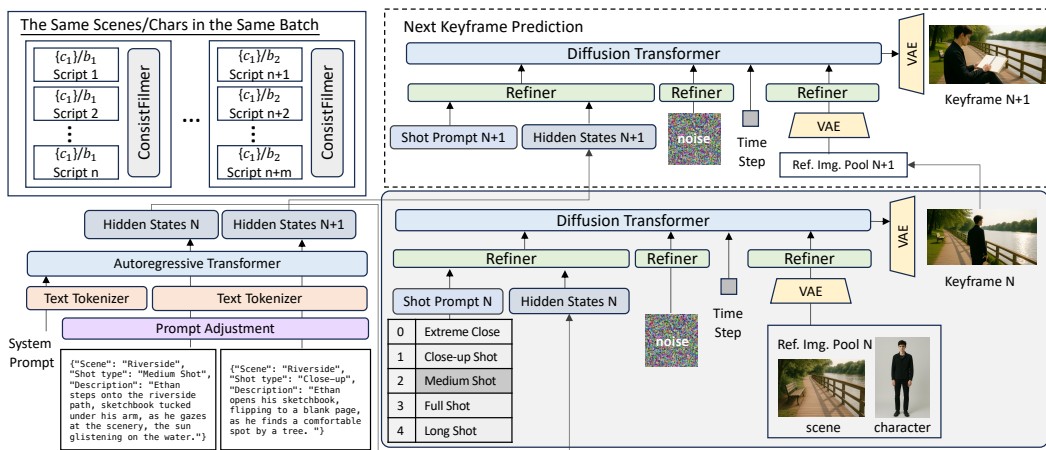

Figure 3: Overview of ConsistFilmer, which can take multimodal materials as input to generate sequences of consistent keyframes.

serve as textual anchors for keyframes. To align textual entities with reference images, we adjust the prompt to the format: $p'_t$ = "In the $\{b_t^{text}\}$ as image1, $\{c_t^{text}\}$ as image2, $\{p_t\}$". This prompt adjustment provides fine-grained grounding in each keyframe and supports multiple characters by extending the mapping.

**Next Keyframe Prediction (NKP).** NKP propagates temporal information across frames. At step $t$, a DiT module synthesizes the current keyframe: $I_t = \text{DiT}(h_t, z_t)$, where $h_t$ is the hidden states from ITR and $z_t$ are VAE-encoded visual features of the reference images $C'_t$ and $b'_t$ in $s_t$. To enhance consistency, the previous frame $I_{t-1}$ is considered to be added to the reference. We resize $I_{t-1}$ by a scale of $\alpha$, re-encoded via VAE and concatenate with reference features: $z_t = [\text{VAE}(C'_t), \text{VAE}(b'_t), \text{VAE}(Scale_\alpha(I_{t-1}))]$ where $\alpha$ denotes the consistency ratio, controlling the strength of conditioning. A larger ratio enforces a stronger signal with past frames, while a smaller one introduces more variation. $\alpha$ trades off the consistency and diversity. This recursive conditioning enables narrative information to flow smoothly across frames.

**Shot-type Control.** Cinematic storytelling requires diverse perspectives. To this end, we introduce a set of shot-type embeddings learned via parameter-efficient prompt tuning on movie data (Bain et al., 2020). This design allows a general-purpose foundation model to better capture the compositional priors needed for generating cinematic keyframes without retraining the entire model. Let $\mathcal{K}$ denote the shot-type vocabulary and $k_t \in \mathcal{K}$ the token assigned to script $s_t$. Given the hidden state $h_t = f_{\text{AR}}(s_t)$, we prepend a learnable shot embedding $E_{\text{shot}}(k) \in \mathbb{R}^{d \times N}$, where $d$ is the embedding dimension and $N$ is the number of tokens, to form a shot-aware representation: $h'_t = [E_{\text{shot}}(k_t); h_t]$. The sequence $h'_t$ is then used to condition the DiT decoder: $I_t = \text{DiT}(h'_t, z_t)$. This hidden-state prefixing enforces shot-specific composition without re-encoding by the AR, leading to outputs that are visually diverse.

**Multi-subject Story Customization.** Prior story generation works (Zhou et al., 2024; Wang et al., 2025b) are restricted to single-character scenarios, limiting their ability to capture multi-character interactions. In contrast, ConsistFilmer conditions on multiple reference images within ITR and NKP, enabling multi-subject story customization and broadening narrative flexibility.

### 3.3 TI2V EXPANSION

Given the prompt text in the scripts from Stage 1 and the keyframes from Stage 2, we employ an existing text-and-image-to-video (TI2V) model (e.g., Veo3 (DeepMind, 2025), Wan (Wan et al., 2025)) to generate short clips. These clips are then concatenated to form long videos, allowing Story2Screen to scale to long horizons while preserving narrative and visual consistency.

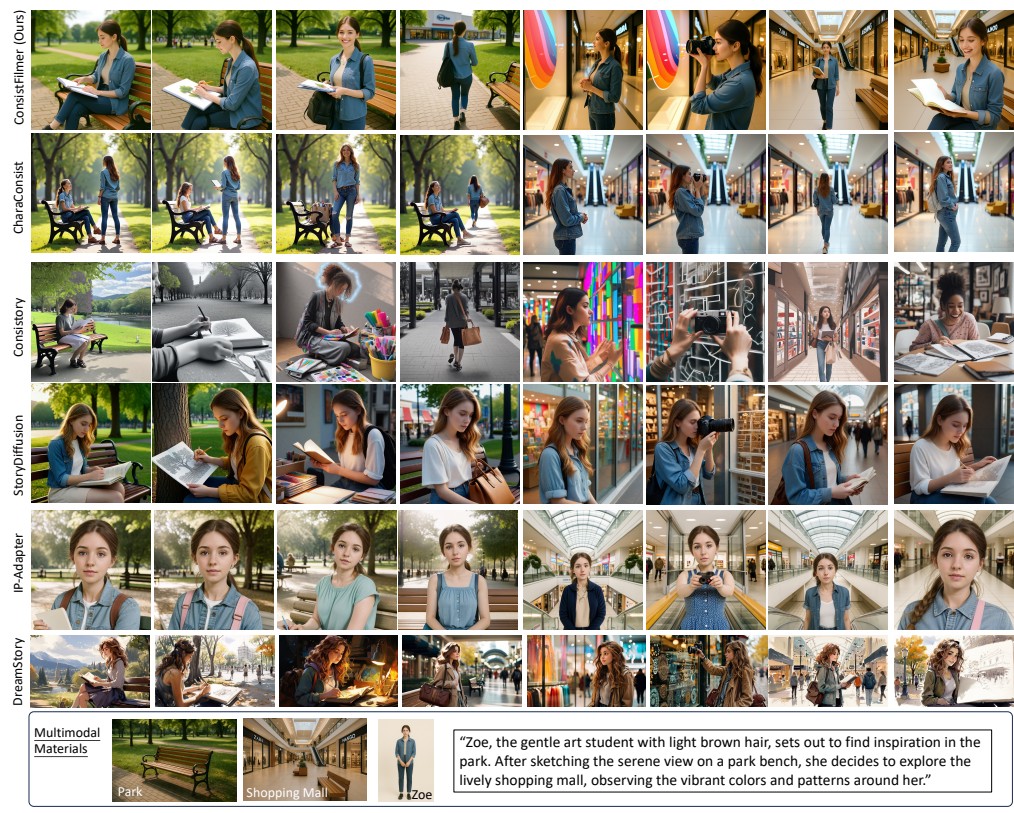

Figure 4: Qualitative Comparison with prior methods on MSB

## 4 EXPERIMENTS

### 4.1 EXPERIMENTAL SETUP

**Implementation Details.** We adopt Omnigen2 (Wu et al., 2025) as backbone to reduce training overhead, which is a common practice in story generation. Omnigen2 integrates an autoregressive model (Qwen2.5-VL-3B) with diffusion transformers. Since our focus is on image generation, we discard the visual inputs in the autoregressive component of Omnigen2. For ConsistFilmer, we optimize the shot-type prompt for 4,000 iterations using four NVIDIA H100 GPUs. During inference, we employ a single NVIDIA 5090 GPU. In NKP, we set $\alpha = 0.75$, and in the shot-type prompt, we set $d = 2048$ and $N = 30$.

**Datasets.** To systematically evaluate the performance of our method, we propose the Multimodal Storyboard Benchmark (MSB) for evaluating single-character story customization and the Multimodal & Multisubject Storyboard Benchmark ($M^2SB$) for assessing multicharacter story customization. The details of the pipeline regarding MSB and $M^2SB$ are provided in the Appendix A.2. Each dataset consists of 100 stories, and each story contains 8 scripts that correspond to 8 keyframes, resulting in 800 Text-and-Image-to-Image (TI2I) instances. In these keyframes, the designated characters appear at the specified locations, ensuring narrative progression. To train the shot-type prompt in the ConsistFilmer, we collect the images from CMD and generate the synthetic data for augmentation. The details of the data preparation pipeline is provided in Appendix A.3.

**Comparison Method.** In the experiments, we compare our method with both training-free and identity-reference approaches. For the training-free category, we select StoryDiffusion (Zhou et al.,

Table 2: Quantitative comparison of prior methods on consistency metrics.

| Method | Inter-Consistency | | Intra-Consistency | | Average Consistency (↑) |
|---|---|---|---|---|---|
| | CLIP-I-fg(↑) | CLIP-I-bg(↑) | CLIP-I-fg(↑) | CLIP-I-bg(↑) | |
| IP-Adapter (Arxiv'23) | 0.901 | 0.936 | 0.900 | 0.646 | 0.846 |
| Consistory (SIGGRAPH'24) | 0.868 | 0.884 | 0.883 | 0.612 | 0.812 |
| StoryDiffusion (NeurIPS'24) | 0.857 | 0.900 | **0.921** | 0.645 | 0.831 |
| DreamStory (TPAMI'25) | 0.844 | 0.858 | 0.896 | 0.635 | 0.808 |
| CharaConsist (ICCV'25) | 0.904 | 0.945 | 0.899 | **0.659** | 0.852 |
| ConsistFilmer (Ours) | **0.905** | **0.961** | 0.914 | 0.657 | **0.858** |

Table 3: Quantitative comparison of prior methods on ID-SIM and image quality/text alignment.

| Method | ID-SIM | | Text Alignment/ Image Quality | | | |
|---|---|---|---|---|---|---|
| | Inter(↑) | Intra(↑) | CLIP-T(↑) | IAS(↑) | IQS(↑) | STA(↑) |
| IP-Adapter (Arxiv'23) | **0.386** | **0.558** | 0.216 | 0.449 | 0.396 | 0.315 |
| Consistory (SIGGRAPH'24) | 0.265 | 0.316 | **0.303** | 0.431 | 0.385 | 0.406 |
| StoryDiffusion (NeurIPS'24) | 0.250 | 0.403 | 0.264 | 0.450 | 0.414 | 0.290 |
| DreamStory (TPAMI'25) | 0.255 | 0.363 | 0.276 | 0.438 | 0.411 | 0.280 |
| CharaConsist (ICCV'25) | 0.303 | 0.400 | 0.265 | 0.448 | 0.415 | 0.247 |
| ConsistFilmer (Ours) | 0.372 | 0.464 | 0.285 | **0.450** | **0.423** | **0.418** |

2024), ConsiStory (Tewel et al., 2024), DreamStory (He et al., 2025), and CharaConsist (Wang et al., 2025b), which generate stories purely from textual descriptions without explicit visual references. These baselines allow us to evaluate the benefit of multimodal conditioning. For the identity-reference category, we adopt IP-Adapter (Ye et al., 2023), a representative method for identity-preserving generation. Following (Wang et al., 2025b), we crop the face region of the reference image using RetinaFace (Deng et al., 2020) before feeding it into IP-Adapter. For models that only accept a single text input, we use the captions of both characters and backgrounds as input. Since the released implementation of DreamStory also supports text-only input, we follow the same protocol.

**Evaluation Metrics.** Following previous works (Wang et al., 2025b), we employ several metrics to evaluate the performance of our multimodal story generation model. CLIP-I measures the pairwise CLIP-based image similarity. To assess consistency more precisely, we compute CLIP-I-fg and CLIP-I-bg, which evaluate the image similarity of foreground and background. We use Dinov2 (Oquab et al., 2023) and Segment Anything Model (Kirillov et al., 2023) to split the foreground and background, and employ Alpha-CLIP (Sun et al., 2024) to compute the similarity in the mask area. To comprehensively evaluate the performance of story customization generation, we further propose inter-consistency and intra-consistency. Inter-consistency represents that we compute the similarity between the reference images and the generated images. As for intra-consistency, in contrast, we compute the similarity between the generated images. We further adopt Identity Similarity (ID-SIM) to measure character consistency across frames. We first use RetinaFace (Deng et al., 2020) to detect the facial area of the character and use FaceNet (Schroff et al., 2015) to extract the face embedding then calculate the similarity between two faces.

## 4.2 EXPERIMENTAL RESULTS

**Qualitative Results.** In Fig. 4 and Fig. 5, we show the results of ConsistFilmer and other methods on single- and multi-subject story customization. As seen in Fig. 4, existing methods struggle to jointly leverage reference images and text, often leading to inconsistencies in backgrounds and character clothing, as in IP-Adapter, StoryDiffusion, and ConsiStory. CharaConsist, while maintaining stable backgrounds, produces scenes that appear overly static. In contrast, ConsistFilmer generates coherent sequences that preserve character identity while ensuring background consistency. In Fig. 5, all comparison methods fail to generate consistent stories with two subjects. As the story progresses, the characters may appear alternately or together, but prior methods that overly emphasize foreground consistency often suffer from character confusion in such cases. In contrast, Story2Screen explicitly determines the required reference images for each frame, including both characters and backgrounds, during the first stage. This design makes it less prone to confusion.

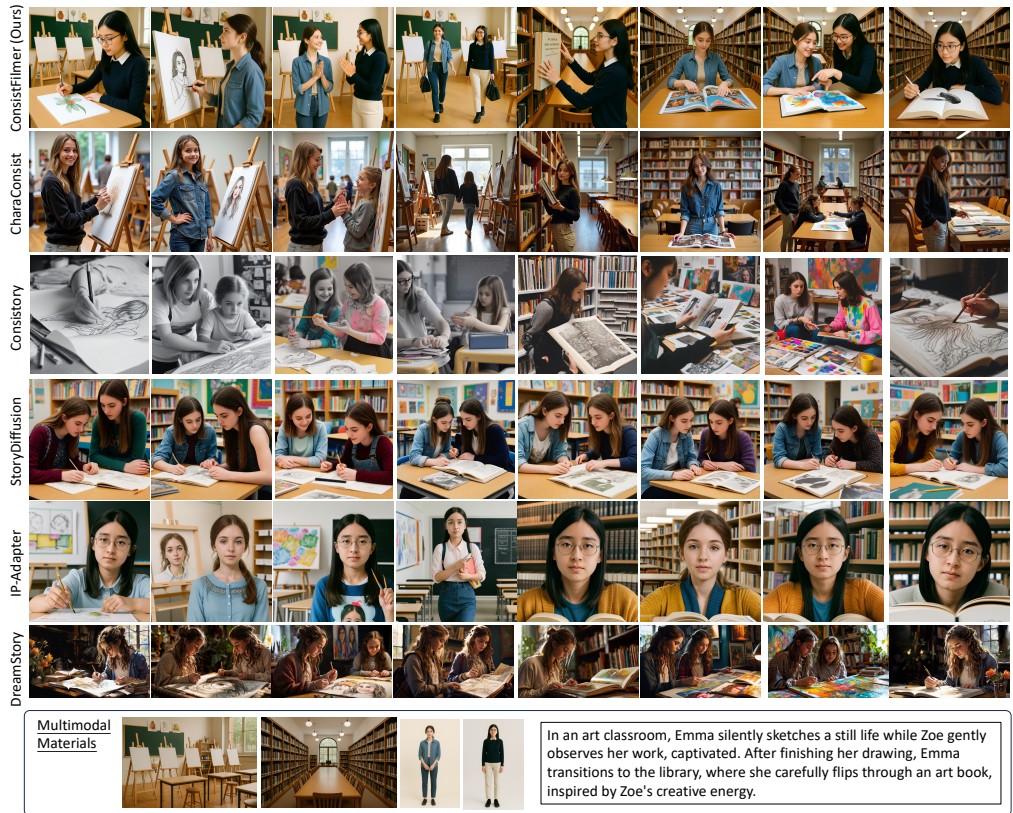

Figure 5: Qualitative Comparison with prior methods on M²SB

Table 4: Comparison of different methods on multiple subject story customization.

| Method | CLIP-T(↑) | IQS(↑) | IAS(↑) |
|---|---|---|---|
| IP-Adapter (Arxiv'23) | 0.249 | 0.487 | 0.392 |
| Consistory (SIGGRAPH'24) | **0.302** | 0.463 | 0.382 |
| StoryDiffusion (NeurIPS'24) | 0.253 | 0.495 | 0.415 |
| CharaConsist (ICCV'25) | 0.263 | 0.492 | 0.424 |
| DreamStory (TPAMI'25) | 0.265 | 0.491 | **0.440** |
| ConsistFilmer (Ours) | 0.272 | **0.496** | 0.439 |

Notably, our method can produce visually richer results by following shot types. More qualitative comparisons and results can be found in Appendix A.5.

**Quantitative Results.** We compare ConsistFilmer with IP-Adapter, StoryDiffusion, ConsiStory, and CharaConsist in Tab. 2 and Tab. 3. ConsistFilmer achieves the best overall consistency, benefiting from image references for stronger inter-consistency. For intra-consistency, its foreground scores are lower due to richer action variations from following textual descriptions, while CharaConsist attains higher background scores by generating nearly identical scenes. This reflects a trade-off between consistency and diversity. In Tab. 3, IP-Adapter shows the highest ID-SIM by producing static, camera-facing characters, whereas ConsistFilmer yields more dynamic and instruction-following results, leading to lower ID-SIM but better story alignment. For text alignment and image quality, our method obtains slightly lower CLIP-T scores compared to ConsiStory, which we at-

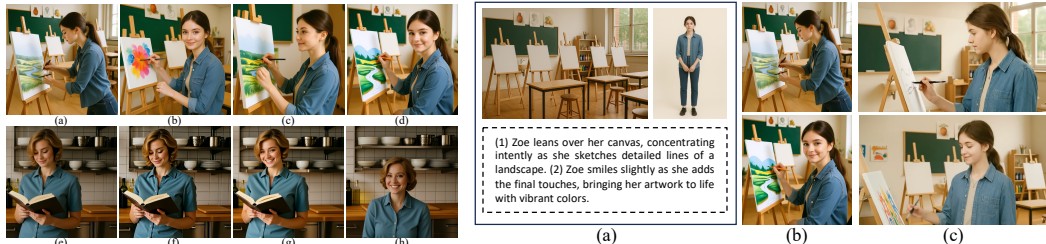

Figure 6: Top: Ablation on consistency components (a) Previous frame, (b) w/o ITR&NKP, (c) +NKP, (d) +ITR&NKP. Bottom: Ablation on $\alpha$, (e) Previous frame, (f) 1, (g) 0.25, (h) 0.125.

Figure 7: Two consecutive frames generated by (b) ConsistFilmer and (c) Veo3 based on the same multimodal input (a).

Table 5: Ablation on ITR and NKP

| ITR | NKP | CLIP-T | Avg-Consistency | ID-SIM |
|-----|-----|--------|-----------------|--------|
| ✗ | ✗ | 0.2834 | 0.854 | 0.316 |
| ✓ | ✗ | 0.2848 | 0.855 | 0.380 |
| ✗ | ✓ | **0.2855** | 0.856 | 0.391 |
| ✓ | ✓ | 0.2852 | **0.858** | **0.418** |

Table 6: Ablation on different $\alpha$

| $\alpha$ | CLIP-T | Avg-Consistency |
|------|--------|-----------------|
| 0.125 | **0.2894** | 0.850 |
| 0.25 | 0.2885 | 0.854 |
| 0.50 | 0.2863 | 0.857 |
| 0.75 | 0.2852 | 0.858 |
| 1.00 | 0.2841 | **0.860** |

tribute to differences in the base models, as also noted in CharaConsist (Wang et al., 2025b). On multi-subject customization (Tab. 4), ConsistFilmer achieves competitive results on M²SB.

### 4.3 ANALYSIS

**Ablation on Consistency Components.** To demonstrate the effectiveness of ITR and NKP, we show quantitative results in Tab. 5. Both components improve consistency, and their combination achieves the best performance. Qualitative examples in Fig. 6 (top row) further show that ConsistFilmer can preserve objects across frames (e.g., the evolving painting) and maintain story flow.

**Ablation on Different Consistent Ratios ($\alpha$).** In Tab. 6, we ablate different consistency ratios, and the corresponding qualitative comparisons are shown in Fig. 6 (bottom row). We observe that increasing $\alpha$ leads to stronger consistency across frames, but at the cost of reduced visual diversity. This is because allocating more tokens to reference images reduces the proportion of text tokens, making the generated results less aligned with textual descriptions and thereby lowering the CLIP-T score. To balance consistency and diversity, we set $\alpha = 0.75$ in our experiments.

**Comparison between ConsistFilmer and Veo.** The Veo series demonstrates strong video generation capabilities with image conditioned synthesis. However, Veo3 generates each frame independently without referencing past outputs, leading to inconsistencies in objects and backgrounds (Fig. 7). In contrast, ConsistFilmer explicitly leverages past keyframes, allowing our framework to maintain narrative continuity and better preserve story progression.

## 5 CONCLUSION

We present Story2Screen, a multimodal framework for story generation. Our method integrates multimodal scripts to generate long visual sequences with both character and scene consistency. To achieve this, we introduced ConsistFilmer, which leverages the power of the multimodal generation model and employs Inner-batch Text Reference (ITR), Next Keyframe Prediction (NKP), and shot-type–aware prompt tuning to produce coherent and cinematic keyframes. We further developed MSB and M²SB, two novel benchmarks that evaluate story generation from consistency, alignment, and shot-type perspectives. Experimental results show that Story2Screen outperforms existing methods in maintaining long-term consistency and enabling controllable story generation. We believe this work provides a step toward more flexible and reliable multimodal story generation, with potential applications in filmmaking, advertising, and storytelling.

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

## A  APPENDIX

### A.1  LARGE LANGUAGE MODEL (LLM) USAGE

This paper benefited from the use of LLM (e.g., ChatGPT) for grammar correction and language polishing. All ideas, experimental designs, and analyses are the sole responsibility of the authors.

### A.2  DETAILS OF MSB & M²SB

The overview of MSB and M²SB is illustrated in Fig. 8. We first use GPT-4o to generate the story outline and the possible characters/scenes images, then we prompt GPT-4o to generate the story scripts in MSB and M²SB for each keyframe, as shown in Fig. 9.

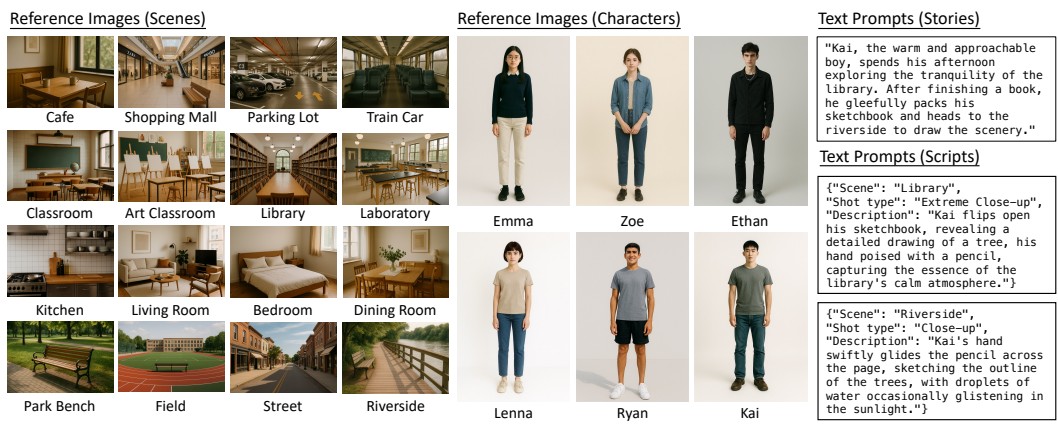

Figure 8: Overview of Multimodal Story Benchmark

### A.3  DATASET FOR SHOT-TYPE CONTROL

In this section, we detail the pipeline used to construct the dataset for shot-type control. (1) We collect video data from the Condensed Movie Dataset (Bain et al., 2020). (2) We apply Byte-Track (Zhang et al., 2022) to track character trajectories across frames, enabling retrieval of the same individual across different scenes. (3) We randomly sample two frames to form a pair and use CLIP (Radford et al., 2021) to verify that both frames depict the same character, thereby avoiding trivial duplication or copy–paste artifacts. (4) We apply a shot-type classifier trained by Xie et al. (2025) to categorize the target frame into one of the canonical shot types. (5) Finally, we use Qwen2.5-VL (Bai et al., 2025) to generate a caption for the target frame, which serves as the textual prompt. The resulting dataset contains 715 example pairs, which we use for training. Examples in this dataset are illustrated in Fig. 10. We further provide more example from Story2Screen on MSB in Fig. 11

### A.4  STORY2SCREEN WITH EXISTING TI2V MODEL.

We employ Veo3 to demonstrate that Story2Screen can be integrated with recent TI2V models to form longer, meaningful videos. Specifically, we leverage the text prompts produced in Stage 1 of Story2Screen and the keyframes generated by ConsistFilmer to synthesize short clips, which are then concatenated into longer videos. We also compare against closed-source models (Sora and Veo3). Story2Screen not only enables the generation of longer videos but also improves text-video alignment with the abstract, resulting in more semantically meaningful content. Qualitative comparisons are provided in Appendix A.4. Story2Screen can generate longer videos with global and local

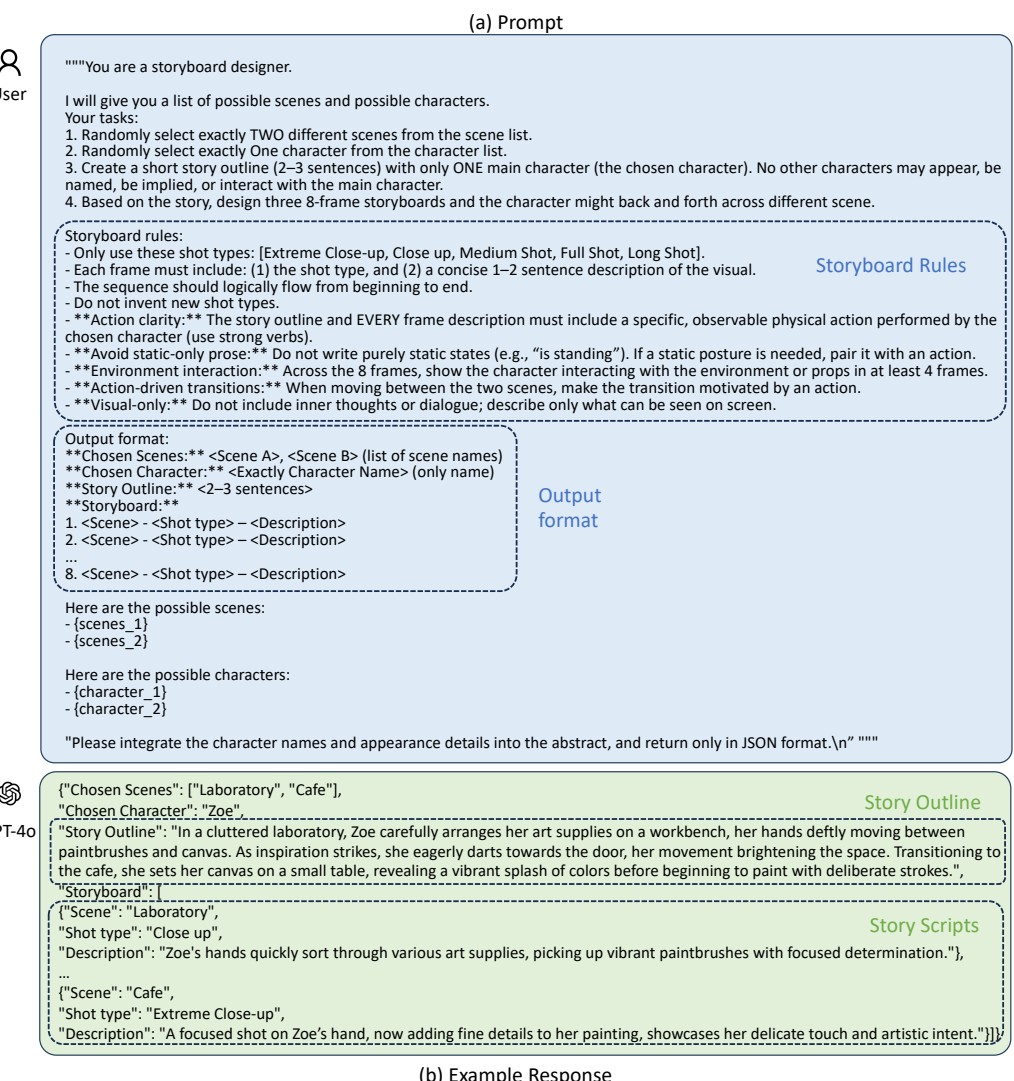

Figure 9: Prompt for GPT to generate scripts prompt in MSB

consistency in the scene, and more diverse shot types. We provide more Qualitative Comparison with the existing T2V model in Figure 12. We also provide the video in the supplementary.

## A.5 MORE QUALITATIVE COMPARISON

We provide more qualitative comparison with existing methods in Fig. 13.

## A.6 LIMITATION

ConsistFilmer primarily relies on the previous frame as a reference to maintain temporal consistency. While this is effective for local continuity, it may be insufficient for capturing long-range narrative structures required in real-world story generation. Future directions could involve integrating higher-level semantic representations, such as multimodal knowledge graphs or narrative planning modules, to provide global guidance and enhance the overall coherence of story progression. We regard this as a promising avenue for future work.

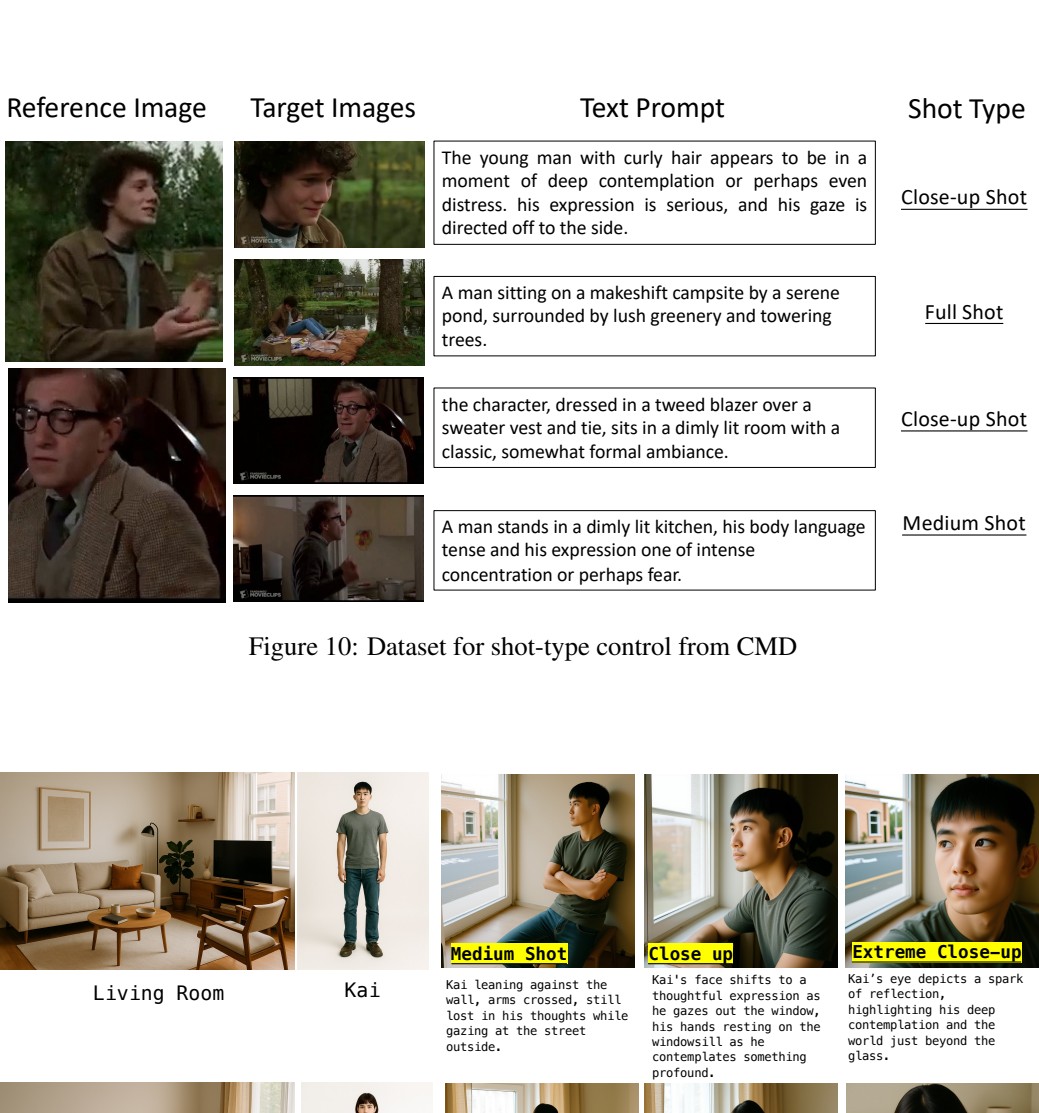

Figure 10: Dataset for shot-type control from CMD

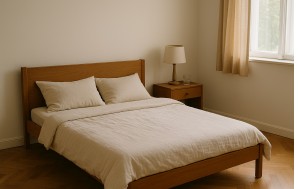
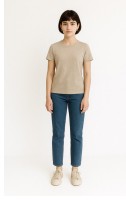
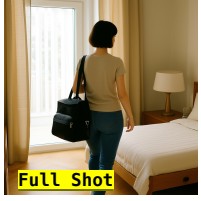
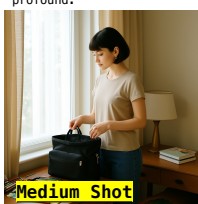
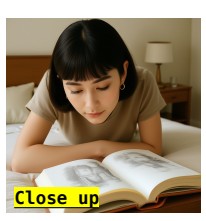

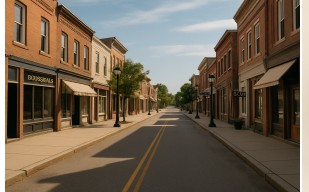
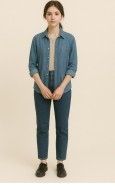
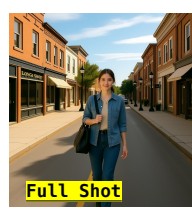
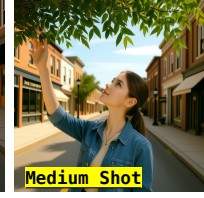
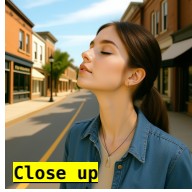

Figure 11: Additional results from MSB

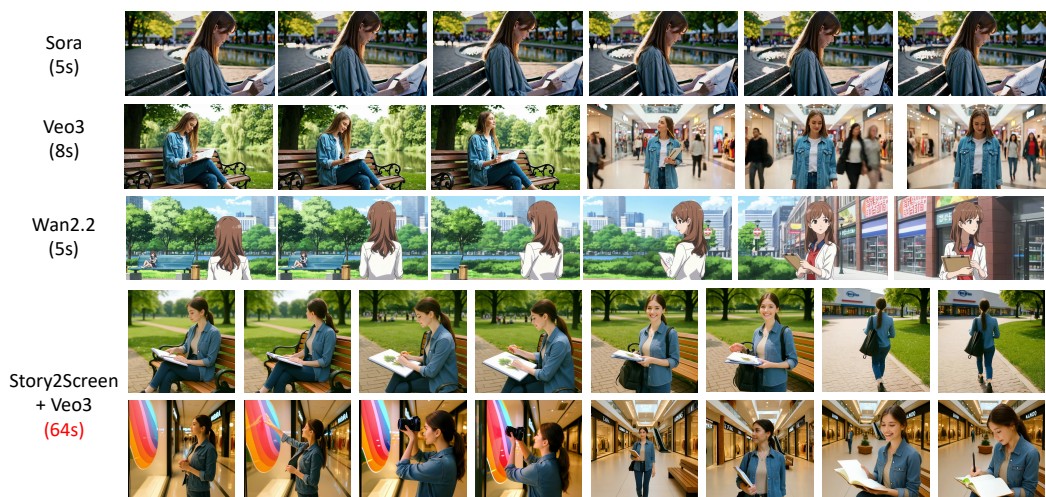

"Zoe, the gentle art student with light brown hair, sets out to find inspiration in the park. After sketching the serene view on a park bench, she decides to explore the lively shopping mall, observing the vibrant colors and patterns around her."

Figure 12: Qualitative Comparison with T2V models.

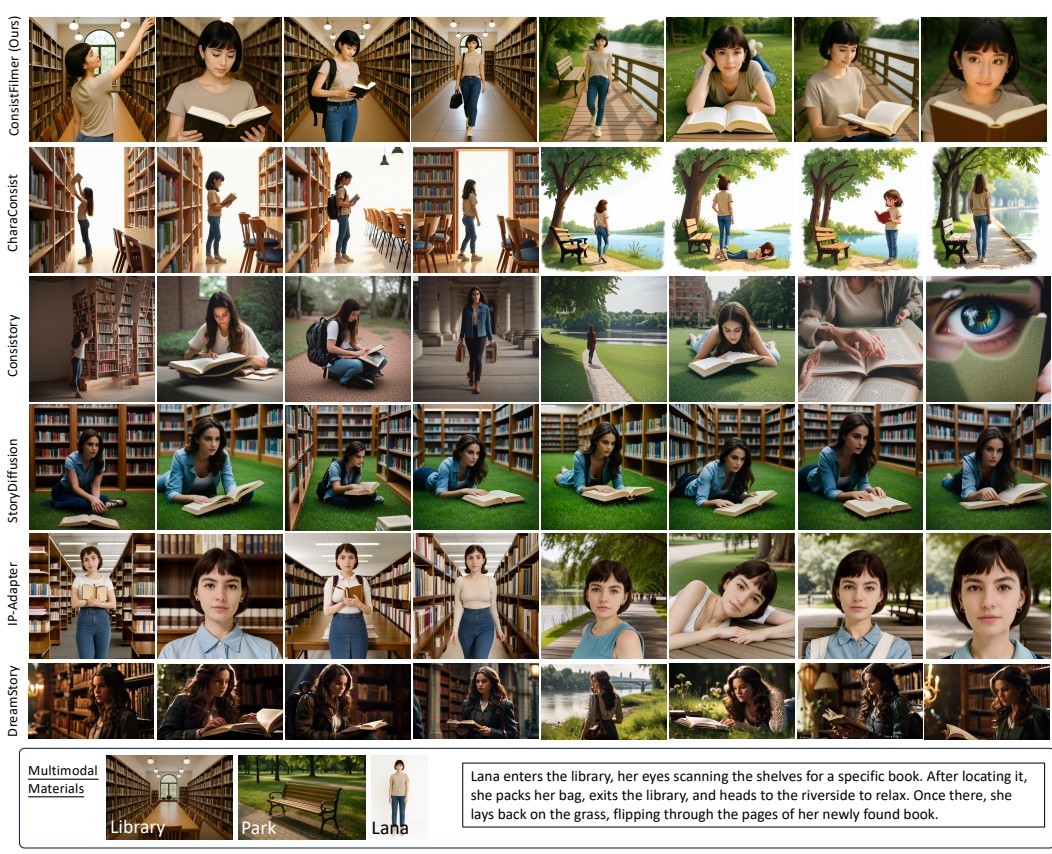

Lana enters the library, her eyes scanning the shelves for a specific book. After locating it, she packs her bag, exits the library, and heads to the riverside to relax. Once there, she lays back on the grass, flipping through the pages of her newly found book.

(1) **[Medium Shot]** Lana stands in front of the towering bookshelves, reaching up to pull down a book from the top shelf.
(2) **[Close Up]** Lana flips open the book, her focused expression reflecting her calm nature as she scans the first few pages.
(3) **[Full Shot]** Lana carefully places the book inside her backpack, adjusting the straps as she prepares to leave.
(4) **[Long Shot]** Lana exits the library, stepping onto the street with a determined look as she starts her journey to the riverside.

(5) **[Long Shot]** Lana approaches the riverside, stepping onto the grass and taking a moment to appreciate the view..
(6) **[Medium Shot]** Lana lays down on the grass, propping herself on one elbow, and opens the book to the first chapter.
(7) **[Medium Shot]** Lana's fingers flip through the pages, her calm demeanor evident as she becomes engrossed in the story.
(8) **[Close-up]** The camera focuses on Lana's eyes as they sparkle with interest, capturing her connection to the book.

Figure 13: Qualitative comparison with prior methods.

