# OpenReview forum: "Story2Screen: Multimodal Story Customization for Long Consistent Visual Sequences"
_ICLR.cc/2026/Conference — ICLR 2026 Conference Withdrawn Submission_

### Official Review · Reviewer_QBMQ · 2025-10-30

**Soundness:** 3
**Presentation:** 3
**Contribution:** 2
**Rating:** 6
**Confidence:** 4

**Summary:**

The paper introduces Story2Screen, a three-stage framework for generating long, consistent visual sequences with multimodal controllability (text, character ID, background, and shot type). Its core component is a keyframe generator named ConsistFilmer, which leverages Inner-batch Text Reference (ITR) and Next Keyframe Prediction (NKP) to enhance consistency. The authors also contribute two new benchmarks, MSB and M$^2$SB, for evaluation.

**Strengths:**

1.  **Task Definition:** The paper defines an interesting task by incorporating shot-type control into the story generation process. This extension is valuable for enhancing the expressiveness of the generated content.

2.  **Component Design:** The proposed ITR and NKP components are direct and effective mechanisms for improving intra-scene and inter-frame consistency, respectively.

3.  **Benchmarks and Results:** The contribution of two new benchmarks (MSB and M$^2$SB) is significant for future standardized evaluation. The qualitative results are strong, demonstrating superior consistency and narrative coherence compared to baselines, particularly in complex scenarios involving scene transitions and multi-character interactions.

**Weaknesses:**

1.  **Limited Methodological Novelty:** The proposed framework, while effective, appears to be a combination of existing ideas. The core components (ITR and NKP) are straightforward implementations of context sharing and autoregressive conditioning. The overall similar to prior work like MovieDreamer (ICLR2025), which somewhat lessens the paper's novelty.

2.  **Short-Range Consistency Mechanism:** The primary mechanism for temporal consistency, NKP, relies solely on the immediate previous frame (`It-1`). This is likely insufficient for maintaining global consistency in very long sequences, especially when handling long-range dependencies (e.g., re-appearance of an object after a long absence). The evaluation, limited to 8-frame sequences, also lacks quantitative evidence to support claims of long-range narrative coherence.

3.  **Experimental Fairness:** The comparison with prior methods is potentially affected by the significant differences in their underlying base models (e.g., OmniGen2 vs. others). It is difficult to ascertain whether the reported marginal improvements on some metrics stem from the proposed methodology or simply from the superior capabilities of the base model itself. This confounding variable impacts the fairness of the comparison.

4.  **Quantitative Evaluation for Cinematic Control:** The paper claims to achieve "cinematic" control via prompt tuning, but this is only supported by qualitative examples. There is no dedicated metric to quantitatively measure the "cinematic quality" or the accuracy of the shot-type control. It would be beneficial if the proposed benchmarks included annotations or evaluation protocols for this dimension.

**Questions:**

My main questions are related to the weaknesses detailed above. In addition, I would like to ask the following:

1.  **Regarding the NKP mechanism:** Have you experimented with conditioning on more than one historical frame (e.g., `It-1`, `It-2`)? Alternatively, have you considered a more global mechanism, such as a  vector that summarizes past keyframes or a cross-frame attention mechanism? Could such an approach enhance long-range consistency without a prohibitive increase in computational cost?

2.  **Regarding the consistency ratio (α):** This ratio creates a trade-off between consistency and diversity. How sensitive is the model's performance to this hyperparameter?

---

### Official Review · Reviewer_khNQ · 2025-10-30

**Soundness:** 3
**Presentation:** 3
**Contribution:** 2
**Rating:** 4
**Confidence:** 3

**Summary:**

The paper introduces Story2Screen, a new framework designed to generate long, coherent visual story sequences with a high degree of customization. The primary goal is to address a key limitation in existing story generation models: the difficulty of simultaneously maintaining consistency for characters, scenes, and textual details when conditioned on diverse multimodal inputs.

**Strengths:**

- The paper does an excellent job of identifying a clear and significant gap in the literature. It convincingly argues that true story customization requires conditioning on more than just text (like CharaConsist ) or just a single character's face (like many ID-preserving methods ).
- The qualitative comparisons in Figures 4 and 5 are striking. Story2Screen produces sequences that are far superior in both character and background consistency compared to all baselines.

**Weaknesses:**

- Relying on LLM for prompt planning is unstable, and once the prompt planning in the first step goes wrong, the error will accumulate. The paper does not explore what happens if a less capable, open-source LLM is used, or how the system handles common LLM failure modes (e.g., logical errors, anachronisms) in the generated scripts.

- The core contribution of this paper lies in generating consistent keyframes. However, the TI2V expansion mainly relies on an existing TI2V model, which avoids the challenge of generating smooth and consistent motion between keyframes.

**Questions:**

- How well does ConsistFilmer handle complex compositions? For instance, a "Long Shot" that is prompted to contain both characters $c_1$ and $c_2$ interacting? The examples in Figure 5 tend to show the two subjects either in different scenes or one after the other.

- How critical is GPT-4o to the success of Stage 1? Have you tested the framework's robustness using less powerful or open-source LLMs (e.g., Llama 3) as the "Director"?

- In the Table 5 ablation, NKP alone seems to provide most of the consistency gains (especially for ID-SIM). Could you provide a qualitative example that specifically isolates the benefit of ITR? For instance, a case where NKP-only fails (e.g., scene context is lost) but ITR+NKP succeeds?

---

### Official Review · Reviewer_45yM · 2025-11-01

**Soundness:** 3
**Presentation:** 3
**Contribution:** 3
**Rating:** 4
**Confidence:** 4

**Summary:**

This paper introduces `Story2Screen`, a three-stage framework designed to generate long, coherent visual sequences (videos) with a high degree of customization. The core problem addressed is the difficulty of existing models in maintaining long-term consistency of characters, scenes, and narrative flow, especially when conditioned on multiple modalities.

The proposed pipeline consists of:
1.  **Multimodal Generative Model as Director:** A large language model (GPT-4o) is used to parse a high-level story description into a sequence of structured multimodal scripts. Each script $s_t$ contains a detailed text prompt $p_t$, references to character images $C'_t$ and a background image $b'_t$, and a desired shot type $k_t$.
2.  **ConsistFilmer:** This is the central contribution, a keyframe generator built upon a diffusion transformer backbone (`Omnigen2`). It introduces three key mechanisms to enhance consistency and control:
    *   **Inner-batch Text Reference (ITR):** Groups and processes text prompts for keyframes within the same scene together to enforce shared context.
    *   **Next Keyframe Prediction (NKP):** Recursively conditions the generation of the current keyframe $I_t$ on the previously generated keyframe $I_{t-1}$, propagating temporal information. This is controlled by a consistency ratio $\\alpha$.
    *   **Shot-type Control:** Uses parameter-efficient prompt tuning to inject shot-type embeddings (e.g., "Medium Shot", "Close-up") into the diffusion model, enabling cinematic diversity.
3.  **TI2V Expansion:** Uses an off-the-shelf Text-and-Image-to-Video (TI2V) model to expand the generated keyframes into short video clips, which are then concatenated.

To evaluate their method, the authors introduce two new benchmarks, **MSB** (Multimodal Storyboard Benchmark) for single-character stories and **M2SB** for multi-character stories. Experiments show that `Story2Screen` (specifically the `ConsistFilmer` component) achieves superior performance in terms of character/scene consistency and text alignment compared to several existing story generation methods.

**Strengths:**

1.  **Comprehensive Multimodal Control:** This is the paper's main strength. It is one of the first works to convincingly demonstrate simultaneous control over textual descriptions, multiple character identities, background scenes, and cinematic shot types within a single framework. This level of fine-grained control is a significant step beyond existing methods.
2.  **Effective Consistency Mechanisms:** The proposed `ITR` and `NKP` mechanisms are simple yet effective. The qualitative and quantitative results, particularly the ablation studies, clearly show their positive impact on both inter-frame and intra-frame consistency. The ability to handle multi-character, multi-scene stories (Fig. 5) is a standout result.
3.  **High-Quality Presentation and Clarity:** The paper is exceptionally well-written and illustrated. The authors do an excellent job of motivating the problem, explaining their solution, and positioning it within the existing literature (Table 1 is a prime example). This makes the paper's contributions easy to understand and appreciate.
4.  **New Benchmarks and Strong Empirical Results:** The introduction of MSB and M2SB provides a dedicated testbed for this complex task. The experimental results are strong, with compelling qualitative examples that clearly highlight the shortcomings of prior work and the advantages of the proposed method.

**Weaknesses:**

1.  **Local-Only Temporal Consistency:** The `NKP` module's reliance on only the *single* previous frame ($I_{t-1}$) is a major architectural limitation. This Markovian assumption is insufficient for ensuring true long-range consistency. For example, a character's clothing detail present in frame 1 might be lost in frame 3 and cannot be recovered in frame 4, as the model only sees frame 3. A more global context mechanism is needed to live up to the promise of "long" sequence generation.
2.  **Fragility of the Pipeline's First Stage:** The entire pipeline is highly dependent on the quality of the scripts generated by GPT-4o. The paper does not discuss the failure modes of this "Director" stage. What happens if the LLM generates illogical plots, inconsistent character actions, or flawed shot-type sequences? The robustness of the entire system to imperfections in Stage 1 is an unaddressed and critical issue.
3.  **Insufficient Evaluation of Final Video Output:** The paper's claim is to generate "long consistent visual sequences" (i.e., videos), but the evaluation is almost entirely focused on the consistency of the generated *keyframes*. The crucial final step—TI2V expansion and concatenation—is not systematically evaluated. For example, there are no metrics for temporal flickering or consistency drops at the boundary between two concatenated clips. The comparison in Figure 7 is anecdotal.
4.  **Benchmark Novelty vs. Generality:** The evaluation is confined to the authors' own benchmarks. While useful, this makes it hard to assess the method's generalizability. A stronger evaluation would have included results on adapted versions of existing video datasets or story datasets to show that the method is not just tailored to the specific structure of MSB/M2SB.
5. **Unfair Comparison due to Backbone Discrepancy:** The main quantitative comparisons are potentially confounded by the choice of backbone models. The proposed method leverages a powerful unified model (Omnigen2), while baselines are based on standard diffusion models. This makes it difficult to isolate the performance gains of the proposed ConsistFilmer techniques from the inherent advantages of the superior backbone. A stronger evaluation would involve comparing methods on the same backbone.

**Questions:**

1.  Regarding the limited temporal scope of NKP: Have you considered or experimented with conditioning on a wider context than just $I_{t-1}$? For instance, using a fixed window of past keyframes (e.g., $I_{t-1}, I_{t-2}$) or incorporating a memory mechanism (like a visual summary token) that aggregates information from all past frames? How does consistency degrade in your current model as the sequence length increases beyond the 8 frames tested (e.g., to 16 or 32 frames)?
2.  Regarding the "LLM as Director": Could you provide an analysis of the robustness of your system to noisy or imperfect scripts from Stage 1? What are the common failure modes of the GPT-4o script generation, and how does `ConsistFilmer` behave when presented with, for example, a script that illogically teleports a character or introduces a continuity error?
3.  Regarding the final video generation (Stage 3): Could you provide a more rigorous evaluation of the final concatenated videos? This could include objective metrics measuring temporal consistency across clip boundaries (e.g., video-based consistency scores) and a user study comparing the perceived quality and coherence of your final videos versus those generated by end-to-end models like Veo3 on the same narrative prompt.
4.  The consistency ratio $\\alpha$ is a fixed hyperparameter. This implies a static trade-off between consistency and diversity for the entire sequence. Have you considered making $\\alpha$ dynamic? For example, it could be predicted by the model or adjusted based on the script (e.g., a low $\\alpha$ for a "jump cut" or scene change, and a high $\\alpha$ for continuous action).
5. Discrepancy in CLIP-T Scores: There is an inconsistency in the reported CLIP-T scores for the final ConsistFilmer model: it is 0.272 in Table 4 but 0.2852 in Table 5. My hypothesis is that Table 4 reports results on the more challenging multi-subject benchmark (M2SB), while Table 5's ablation study was conducted on the single-subject benchmark (MSB). Could the authors confirm this and explicitly state the dataset used for the ablation study in the paper to avoid confusion? A brief discussion on why the multi-subject task leads to a slightly lower text-alignment score would also strengthen the analysis.

---

### Official Review · Reviewer_Ugvu · 2025-11-01

**Soundness:** 3
**Presentation:** 2
**Contribution:** 2
**Rating:** 2
**Confidence:** 4

**Summary:**

This paper aims to generate story videos conditioned on multiple texts and images. It is a three-stage pipeline. Stage-I uses GPT4o to produce multimodal scripts (text prompts, shot types, and selected reference images). Stage-II utilizes the proposed ConsistFilmer to generate consistent keyframes, which is supported by the introduced Inner-batch Text Reference and Next-Keyframe Prediction modules. Stage-III expands those keyframes into video clips using external text-image-to-video models (Veo3/Wan). The proposed method is compared with recent methods (CharaConsist, DreamStory, StoryDiffusion, Consistory, and IP-Adapter) on the two newly constructed benchmarks and outperforms them across most metrics. This paper also presents some ablation studies on the choices of consistent ratios and consistency components.

**Strengths:**

1. The exploration of the long-horizon, identity- and scene-consistent story visualization problem is well-motivated and invaluable.
2. Shot-type control via lightweight prompt-tuning is practical and aligns with professional film production.
3. The introduced MSB and M2SB provide fixed scripts, identities, and backgrounds for repeatable comparisons.

**Weaknesses:**

1. The work is largely an incremental application built on OmniGen2, with marginal methodological innovation.
2. NKP is a very straightforward frame-prediction heuristic, and ITR brings only marginal improvements in CLIP-T / avg-consistency, while the larger ID-SIM gain seems unconvincing given it only adjusts hidden text states.
3. Benchmark diversity is low. Both image and text data are limited and generated by GPT models, which restricts the benchmarking domain.
4. There is no human evaluation to confirm whether the generated stories are actually preferred or perceived as more consistent.
5. In the demo video (around 36s), noticeable identity drift appears: the Western woman's face reference turns into an East Asian face midway.
6. The current clip-stitching pipeline (i.e., the 3rd stage) cannot maintain consistency of newly generated content across segments.

**Questions:**

Different baselines rely on different base models and reference conditioning. How can you ensure fairness among the comparisons?

---

### Note · Authors · 2025-11-14

I have read and agree with the venue's withdrawal policy on behalf of myself and my co-authors.